# A Knowledge-Graph-Based Multimodal Deep Learning Framework for Identifying Drug–Drug Interactions

**DOI:** 10.3390/molecules28031490

**Published:** 2023-02-03

**Authors:** Jing Zhang, Meng Chen, Jie Liu, Dongdong Peng, Zong Dai, Xiaoyong Zou, Zhanchao Li

**Affiliations:** 1School of Chemistry and Chemical Engineering, Guangdong Pharmaceutical University, Guangzhou 510006, China; 2School of Biomedical Engineering, Sun Yat-sen University, Guangzhou 510275, China; 3School of Chemistry, Sun Yat-sen University, Guangzhou 510275, China; 4Key Laboratory of Digital Quality Evaluation of Traditional Chinese Medicine, National Administration of Traditional Chinese Medicine, Guangzhou 510006, China

**Keywords:** drug-drug interaction prediction, knowledge graph convolutional networks, neural factorization machines

## Abstract

The identification of drug–drug interactions (DDIs) plays a crucial role in various areas of drug development. In this study, a deep learning framework (KGCN_NFM) is presented to recognize DDIs using coupling knowledge graph convolutional networks (KGCNs) with neural factorization machines (NFMs). A KGCN is used to learn the embedding representation containing high-order structural information and semantic information in the knowledge graph (KG). The embedding and the Morgan molecular fingerprint of drugs are then used as input of NFMs to predict DDIs. The performance and effectiveness of the current method have been evaluated and confirmed based on the two real-world datasets with different sizes, and the results demonstrate that KGCN_NFM outperforms the state-of-the-art algorithms. Moreover, the identified interactions between topotecan and dantron by KGCN_NFM were validated through MTT assays, apoptosis experiments, cell cycle analysis, and molecular docking. Our study shows that the combination therapy of the two drugs exerts a synergistic anticancer effect, which provides an effective treatment strategy against lung carcinoma. These results reveal that KGCN_NFM is a valuable tool for integrating heterogeneous information to identify potential DDIs.

## 1. Introduction

Drug–drug interactions (DDIs) refer to the phenomenon whereby one drug alters the pharmacological or clinical responses of another co-administered drug. Inappropriate drug combinations may lead to adverse drug reactions and may even be life-threatening [1]. It is estimated that 30% of adverse side effects are caused by DDIs [2], leading to astronomical medical costs as well as substantial morbidity and mortality [3,4], which is the main reason for the withdrawal of drugs from the market [5]. However, traditional experimental methods for detecting DDIs are usually inefficient, expensive, and time-consuming [6,7]. Therefore, it is urgent to develop a more precise computational method for distinguishing potential DDIs.

In the past decades, various machine learning methods have been proposed [8,9,10,11] to discriminate the DDIs. The vast majority of these methods are based on the assumption that if drug A and drug B produce a specific biological effect through their interaction with each other, then drug C similar to drug A is likely to interact with drug B and performs the same effect. For instance, Zhang et al. [12] proposed an ensemble approach for DDI prediction, in which 14 kinds of drug–drug similarities were derived from chemical, biological, phenotypic, and topological data. Zhang et al. [13] developed a matrix completion method, where eight kinds of drug–drug similarities were considered as a part of regularization for conserving drugs manifold in lower space. Gottlieb et al. [14] introduced an inferring drug interaction (INDI) method that considered seven drug–drug similarities to infer the pharmacokinetic (PK) and pharmacodynamic (PD) DDIs. Although these methods have been successful, they still have some difficulties in dealing with complex data structures such as drug interaction networks.

Several methods were further proposed to overcome these limitations by mining knowledge from heterogeneous networks [15]. Nodes in these networks represent drugs, proteins, or other biological entities, while edges mean the interactions, associations, or similarities between these nodes [16,17]. Most of these methods employ network embedding algorithms such as DeepWalk [18], GraphWave [19], and GraRep [20] to capture complex network topology information. Nevertheless, these models ignore the characteristics of nodes and the types of edges and focus only on the connections between nodes [21]. Hence, knowledge graph (KG)-based methods have been given more attention and successfully utilized to distinguish DDIs. KG embedding techniques can be roughly divided into two categories: translational distance models and semantic matching models [22]. The first class includes TansE [23], TransH [24], and TransR [25], in which distance represents rationality between entities. For instance, Abdelaziz et al. [26] improved the prediction performance by using TransH and HolE [27] to learn the embedding of drugs and relations, then combined those with other features (such as enzymes, chemical substructures, and pathways). The second class performs matching latent semantics and relation. For example, Zitnik et al. [28] utilized RESCAL [29] to identify the most appropriate relation between drugs by considering each relation as a matrix. Though these approaches are promising, they usually only focus on the exploration of drug-related triples, not on the properties of the drug. Therefore, it is necessary to integrate drug attributes into the representation learning process to improve the performance of DDI prediction.

Different from the multi-relational embedding methods mentioned above, network structure-based embedding methods focus more on the network topology, which is used in learning entity embedding by aggregating the information of their neighbors. With the growing popularity of graph neural networks (GNNs), including graph convolution networks (GCNs) and graph attention networks (GATs) [30], many graph-based methods have been successfully used to predict DDIs. For example, Feng et al. [31] proposed a two-layer GCN for learning node embedding, then adopted a deep neural network to identify potential DDIs. Zitnik et al. [32] developed a multimodal graph to predict side effects by modeling DDI identification as a multi-relational link prediction problem. However, these methods only consider the attributes of nodes and/or neighbors and the structural characteristics of the graph. In fact, learning entity embedding can be affected by multiple relationships and network structure information. In this regard, a KGCN can dispose of the problem by combining the idea of the GCN to create higher-order structures and semantic relations [33]. In addition, NFMs is a recommendation-based model and has been widely used in web-based e-commerce [34] and drug–target interaction prediction [35]. It effectively combines the linearity of factorization machines (FMs) in modeling second-order feature interactions and the nonlinearity of neural networks in modeling higher-order feature interactions [36]. An NFMs captures the lower-level second-order feature interactions through a feature cross-pooling layer and provides more effective information than directly concatenating features.

In this study, a model called KGCN_NFM was proposed to recognize DDIs by incorporating KGCNs and NFMs. KGCN_NFM can be regarded as a pre-training model based on a knowledge graph, and its features are transferred to the downstream tailor-made recommendation system. A KGCN is utilized to capture latent semantic and structural information in knowledge graphs. The low-dimensional embedding representations and Morgan molecular fingerprints (MorganFP) of drugs are combined by using a recommendation-based NFMs. Furthermore, the new DDIs predicted by KGCN_NFM between topotecan and dantron was validated through MTT assays, apoptosis experiments, cell cycle analysis, and molecular docking, and we demonstrated the potential novel applications using two drugs for lung carcinoma. All these results demonstrate that our method can provide a useful tool to predict unknown DDIs. The schematic workflow of the current method is illustrated in Figure 1.

## 2. Results and Discussion

### 2.1. Effect of Threshold on Model Performance

Based on four thresholds 0.6 AED, 0.8 AED, 1.0 AED, and 1.2 AED, a series of reliable negative sample datasets were constructed to examine their impact on the performance of the model. Moreover, the random selection strategy of negative samples was also used to construct negative sample datasets in order to reveal the effectiveness of the current strategy. As shown in Table 1, the performance of the model gradually improves with the threshold increase. At a 0.6 AED threshold, Acc, Sen, Spe, Pre, F1, MCC, AUROC, and AUPRC were 96.40%, 97.07%, 95.74%, 95.80%, 0.9643, 0.9282, 0.9917, and 0.9895 respectively, about 2.24%, 1.58%, 2.90%, 2.84%, 0.0221, 0.0447, 0.0075, and 0.0097 lower than those derived from the threshold 1.2 AED. Furthermore, the performances of the models from the reliable negative sample selection were consistently higher than those from the random selection strategy, suggesting that the current negative sample selection strategy has a positive impact on the prediction performance. Finally, 1.2 AED was selected as a threshold for the reliable negative sample selection and mode construction.

### 2.2. Optimization of Model Parameter

As shown in Figure 2A, experiments were first conducted by varying the learning rate of the NFMs (0.0005, 0.001, 0.005, 0.01, and 0.02). The performance of the current method was gradually improved with the increase of *lr* and achieved the optimized result when *lr* was equal to 0.005, and then the performance was degraded. Then the influence of neighbor size *N* was investigated by varying it from 3, 5, 7, 9 (Figure 2B). Obviously, all performance metrics were improved as the number of neighbors was increased, and the best performance was obtained when the neighbor size was set to 7. Subsequently, it reached a plateau phase and the performance remains stable. The results were caused by the fact that too small a neighbor size has insufficient capacity to incorporate neighborhood information, while an overlarge neighbor size is prone to be misled by noises [37]. Intuitively, the model with two layers achieved a consistent improvement over that with one layer across all the metrics, while the performance suffers from degrading when the layer size is higher than 2 (Figure 2C). In addition, the influence of embedding dimension *d* was also examined by varying it from 16 to 128. According to Figure 2D, KGCN_NFM performs better with the increase of dimensions, indicating that the performance can be enhanced with a proper *d* that can encode enough information about drugs and entities from the KG. However, the model may suffer from overfitting when *d* is too large. For distinguishing the type of interaction between two drugs, it is also investigated how the batch size affects the performance. From Figure 2E, KGCN_NFM can achieve the best performance when the batch size is set to 1250.

### 2.3. Comparison of Various Embedding and Fingerprint Features

In this section, various embedding and fingerprint descriptors were compared to obtain the optimized performance. As illustrated in Figure 3A, the EstateFP descriptor achieved the lowest average Acc, Sen, Spe, Pre, F1, MCC, AUROC, and AUPRC, which may be caused by the fact that the descriptor is only a 79-dimensional feature vector and cannot sufficiently capture the drug structure information. For AP2D, MACCS, PubFP, and SubFP, the results of Acc from the 10-fold CV were about 97.84%, 97.86%, 97.98%, and 98.64%, about 0.91%, 0.89%, 0.77%, and 0.11% lower than those of MorganFP, respectively. Although the training time of SubFP is shorter than that of MorganFP (Figure 3B), the RSD of the former is larger than that of the latter, indicating that the prediction model constructed by MorganFP is more robust. Therefore, the MorganFP descriptor is the feasible feature for characterizing drug structure in the current study.

As shown in Figure 3C, KGCN was almost the same as the traditional network representation learning methods DeepWalk, GraphWave, and GraRep based on the Karate Club [38]. Although the Acc of KGCN_NFM was 0.2% lower than that of DeepWalk, it seems to be insignificant compared to the training time of prediction in the NFMs model (Figure 3D), because the KGCN is 15 times faster than DeepWalk. From the perspective of time cost, Morgan molecular fingerprints took the least training time. The KGCN can save a lot of computing resources since it is able to take advantage of more sophisticated structural and semantic features of the KG and integrate them into the embedding information, while the other methods only consider the topological properties of the network. Therefore, it can be concluded that the incorporation of additional neighbor information into the embedding process plays a key role in enhancing the performance of the model.

### 2.4. Comparison with Other Methods

A comparison between KGCN_NFM with five baseline methods, including RF, KGNN, CNN-LSTM, KGE_NFM, and DDKG, was performed based on the constructed datasets. From Table 2, KGCN_NFM consistently yields the best performance on both two datasets. For the benchmark dataset, the RF model has the worst performance. KGCN_NFM improved over the strongest baseline KGNN model with respect to Acc by 1.68%, Sen by 0.27%, Spe by 3.06%, Pre by 2.89%, F1 by 0.0016, MCC by 0.0327, AUC by 0.0069, and AUPR by 0.0083. In addition, our method has the lowest RSD (%) compared with other methods. For discriminating types of increase or decrease, our model also achieved good performance. These results demonstrate the excellent performance of the current method. For the KEGG-DDI dataset, KGE-NFM performs the worst, which may be caused by focusing on the semantic relationship of drugs in the KG, ignoring the network topology, and leading to poor generalization. In contrast, our model compensates for this deficiency and achieves excellent performance.

Table 3 summarizes the performance results in real scenarios. A significant degradation in performance was found in the PW-CV scenarios. The PW-CV split is more challenging to the DDI prediction models compared to the traditional 10-fold CV, which is consistent with the fact that the PW-CV split can prevent the feature information of drugs from leaking into the test set. Although DDKG achieved the highest Acc of 86.74%, Sen of 97.13%, and F1 of 0.8815, Spe was only 76.03% and was lower than 80%, revealing that the model may lead to a high false positive rate (i.e., more negative samples are predicted as positive samples). In contrast, our method obtained the highest Pre of 84.39%, AUC of 0.9083, and AUPR of 0.8775. More importantly, Sen and Spe were 82.45% and 83.95%, respectively. These two values are close to each other, indicating that the prediction results of the current model for positive samples and negative samples are not skewed. Meanwhile, we can also notice that the recognition results of all other methods are skewed for either positive samples or negative samples. In addition, the Acc obtained by our method is the second highest, and about only 3% lower than that of DDKG. At the same time, it is about 11%, 16%, 32%, and 44% higher than those of KGE_NFM, RF, CNN-LSTM, and KGNN, respectively. All these results show that the current method is still optimal for PW-CV scenarios.

### 2.5. Ablation Study

To determine whether the components of our method play an important role in the prediction performance, an ablation study was conducted by removing the KGCN or NFMs module. As shown in Figure 4, KGCN_NFM has better performance than separate modules using the NFMs or KGCN. The performance of the model without KGCN (AUROC = 0.9912, AUPR = 0.9373) is slightly lower than those of the overall framework (AUROC = 0.9992, AUPR = 0.9992), but it confronts the problem of long training time. There is no doubt that the embedding learned by the KGCN is efficient because it utilizes the multi-relational information between drugs and other entities as well as the structure information of the KG to provide a fast-learning, effective representation for the reduction of training time. In addition, good performance was obtained after removing the NFMs module (AUROC = 0.9860, AUPR = 0.9435), but it was still low compared with the whole. The reason may be caused by the limited amount of feature information captured by the KGCN. Hence, the NFMs successfully makes up for the shortcomings of the KGCN by performing high-order feature interactions. In general, the two modules perform their respective functions, and combining them together can mine richer feature information and ultimately improve their performance.

### 2.6. Case Study

Case studies were conducted to verify the reliability of KGCN_NFM in practice. The top-ranked 10 predicted drugs were focused related to topotecan, which has a single target and usually overcomes its resistance through a multidrug combination [39]. These predictions were validated by the drug.com Interaction Checker tool or the literature and listed in Table 4. For example, topotecan and diroximel fumarate were predicted to have interactions, and their combination increases the risk of serious infections according to drugs.com (https://www.drugs.com/drug-interactions/dimethyl-fumarate-with-topotecan-3464-0-2217-0.html (accessed on 1 January 2023)).

The cytotoxicity and drug combination effects of recognized drug pairs were evaluated through an MTT assay against non-small cell lung cancer (A549). As can be seen in Figure 5A, both topotecan and dantron showed significant cytotoxicity against A549 cells in a concentration-dependent manner. Their IC_50_ values were also calculated to be 26.41 ± 3.44 μM and 52.36 ± 6.59 μM, respectively. The cytotoxicity of topotecan increased significantly when incubated for 48 h at 37 °C with dantron (Figure 5B). Based on the combination analyses of the data (Figure 5C), topotecan and dantron showed synergistic effects (CI values < 1) in A549 cells. These experimental results match the predictions of our model, revealing that our method is reliable, persuasive, and expected to become a practical drug screening tool.

According to Figure 6A, the concentration of 10 μM dantron had no effects on cell apoptosis. However, early apoptosis (Q3) and late apoptosis (Q2) were observed in both the topotecan monotherapy and combination groups. In combination groups, both early and late apoptosis were more pronounced than in the topotecan monotherapy group, revealing that dantron increased topotecan-induced apoptosis in A549 cells. As shown in Figure 6B, the concentration of 10 μM dantron had no effects on cell cycle arrest, but A549 cells were arrested in the S phase after 48 h with 5 μM topotecan. Presumably, because the DNA inside the cell was damaged, the cell cycle was arrested [40]. We also found that the cell cycle was arrested after 48 h of topotecan combination treatment. Compared with the topotecan monotherapy group, cells in the S phase were increased in the combination group, indicating that the combination therapy could enhance the cell cycle-arresting effect of topotecan.

Previous studies have given evidence that topotecan treatment upregulated the expression of sestrin 2, induced the phosphorylation of the AMPKα subunit at Thr172, and inhibited the mTORC1 pathway [41]. Zhou’s research also found that dantron lipid metabolism by activating AMPK [42]. Therefore, dantron and topotecan were docked with key proteins (AMPK, mTOR, p53, sestrin, ULK1) in the AMPK/mTOR pathway by Autodock software. The results of docking are shown in Figure 7, The dantron–mTOR complex and topotecan–AMPK complex had the lowest binding energies, which are −5.6 kcal/mol and −5.78 kcal/mol, respectively. Additionally, their inhibition constants were 78.32 μM and 57.92 μM, respectively. The non-bonding interactions between the two ligands and protein complexes are mainly hydrogen-bonding interactions and hydrophobic interactions (Figure 7). These results suggest that dantron and topotecan may act on different proteins in the AMPK/mTOR pathway and exert synergistic antitumor effects.

## 3. Materials and Methods

### 3.1. Construction of Knowledge Graph and Dataset

The DDI pairs, drug–target association pairs, and drug–enzyme association pairs were parsed and extracted from the XML file (Version 5.1.8) downloaded from DrugBank [43]. A total of 1,113,431 DDIs were obtained and considered positive samples of the benchmark dataset. These collected DDIs were divided into 446,686 increased DDIs and 666,745 decreased DDIs. Increased and decreased mean that a DDI can increase or decrease the effects of one or both drugs. The information on drug–gene associations, drug–disease associations, and gene–disease associations were downloaded from the CTD database [44]. The protein–disease relationships were obtained from the Uniprot database [45]. The protein–protein interactions were collected from the HIPPIE database [46]. Then, entity alignment, simplification, and deletion of irrelevant items were performed (for details, see Appendix A). Then multi-level information including drugs, proteins, genes, diseases, and eight relationships were stored in the KG in the form of triples. Each of the triples represents an interaction or association between two entities (such as drug, drug–target association, target). Finally, the constructed KG contains 7129 drugs, 15,184 proteins, 8083 genes, 5579 diseases, and 1,381,589 triples. The triples include 1,113,431 DDIs, 15,190 drug–target associations, 5085 drug–enzyme associations, 9310 drug–gene associations, 64,879 drug–disease associations, 16,634 gene–disease associations, 150,338 protein–protein interactions, and 6722 protein–disease associations. To further evaluate the performance of our model, the KEGG-drug dataset containing 55,631 DDIs among 1583 drugs was also constructed [47].

### 3.2. Characterization of Drug–Drug Interactions

Fingerprint features from molecular structure and embedding from KG were employed to characterize DDIs. The MorganFPs (1024 bits) were calculated by using RDKit [48]. Meanwhile, molecular fingerprint descriptors of MACCS, Estate (EstateFP), substructure (SubFP), PubChem (PubFP), and 2D atom pair (AP2D) were also calculated by using PaDEL-Dsecriptor [49] to compare the performance of various fingerprint features. The KGCN model was utilized to learn potential entity embedding representations for all entities. The embedding representation contains the network relationship and semantic information on drugs in the KG, and a feature vector with 64 dimensions was adopted to characterize the drugs. Finally, a drug–drug interaction pair can be described as a feature vector with 2176 dimensions by concatenating the molecular fingerprint features (1024 dimensions) and graph embedding features (64 dimensions) of two drug molecules.

### 3.3. Selection of Reliable Negative Sample

Considering that unlabeled samples may contain potential positive samples (i.e., DDIs that have not been verified by experiments), randomly selecting negative samples from unlabeled ones may affect the prediction performance of the model. Therefore, the positive unlabeled learning method proposed by Li et al. [50] was adopted to select reliable negative samples. Steps are as follows: (1) use the fingerprint feature of the drug to calculate the mean of each dimension in the positive samples and form a 2048-dimensional feature vector (i.e., cluster center); (2) calculate the Euclidean distance between all unlabeled samples and the cluster center, and the average Euclidean distance (AED); (3) set a threshold D (D = n × AED, n ∈ ℝ^+^), and an unlabeled sample can be regarded as a reliable negative sample if the distance from the unlabeled one to the cluster center is higher than the threshold.

### 3.4. Construction of Identification Model

As shown in Figure 1, the KGCN_NFM model consists of two main components. (1) Utilizing the KGCN model to sample from the neighbors of each entity in KG as their receptive field. Then combine neighborhood information with bias when calculating the representation of a given entity. (2) Using NFMs and DNN to distinguish potential DDIs from unknown and unlabeled ones.

#### 3.4.1. KGCN Layer

The KGCN model comprises two components: (i) message propagation and (ii) message aggregation. In the first part, suppose that there is a candidate drug pair *d**_i_*–*d_j_*, where *i*, *j* ∈ ℝ^d^, and *i* ≠ *j*. *N_neigh_* represents the set of neighbor entities directly connected to *d**_i_* and *N_neigh_* ⊂ *N_e_*, *N_e_* means all entities involved in KG. ➀ Calculate the score between *d**_i_* and relation *r* using Equation (1):(1)$$g\left(i,r\right)=\pi_{r}^{i}$$
where *i* and *r* represent drug *i* and relation *r*, respectively. $\pi_{r}^{i}$ refers to the inner product performance: ℝ^d^ × ℝ^d^ → ℝ, which means the correlation of a relation *r* to a drug *i*. ➁ ${\tilde{\pi }}_{r_{N_{neigh}}}^{i}$ is the normalized drug-relation score and calculated with Equation (2):(2)$${\tilde{\pi }}_{r_{N_{neigh}}}^{i}=\frac{exp\left(\pi_{r_{neigh_{\left(e\right)}}}^{i}\right)}{\underset{e\in N_{neigh}}{\sum }exp\left(\pi_{r_{neigh_{\left(e\right)}}}^{i}\right)}$$
where *e* denotes the representation of entity *e*. ➂ The linear combination of the neighborhood of drug *j* is computed to describe the topological proximity structure of entity *e* and calculated with Equation (3):(3)$$e_{N_{neigh}}^{i}=\underset{e\in N_{neigh}}{\sum }{\tilde{\pi }}_{r_{N_{neigh}}}^{i}e$$

Instead of using a full set of neighbors $N_{neigh}$, a fixed size set *S*(v) is uniformly sampled considering the efficiency and the fixed computation pattern of each batch. Therefore, the neighborhood representation of entity *e*, $e_{S\left(v\right)}^{i}$ can be obtained.

The second part aggregates the entity embedding representation *e* and its neighborhood representation $e_{S\left(v\right)}^{i}$ into a single vector in a KGCN layer. There are three types of aggregators (*Agg*): 

The sum aggregator takes the summation of two embedding representations before applying nonlinear transformation and operation using Equation (4):(4)$$Agg_{sum}=\sigma \left(W_{\mathrm{sum}}\left(e+e_{S\left(v\right)}^{i}\right)+b_{\mathrm{sum}}\right)$$

The concat aggregator [51] concatenates the two embedding representations and adopts a nonlinear transformation using Equation (5):(5)$$Agg_{concat\;}=\sigma \left(W_{\mathrm{concat}}\left(e\;\|\;e_{S\left(v\right)}^{i}\right)+b_{\mathrm{concat}}\right)$$

The neighbor aggregator only takes the neighbor embedding representation of entity v as the output representation, followed by a nonlinear transformation using Equation (6):(6)$$Agg_{neigh\;}=\sigma \left(W_{\mathrm{concat}}\cdot \;e_{S\left(v\right)}^{i}+b_{\mathrm{neigh}}\right)$$

In these equations, σ denotes the activation functions such as ReLU [52] and *W* and *b* are trainable parameters. We evaluated these aggregators for details in Appendix A, and $Agg_{concat\;}$ was adopted in the KGCN model.

#### 3.4.2. NFMs and DNN layer

An NFMs is constructed to identify potential DDIs. The scoring function of the NFMs is described as Equation (7):(7)$$\hat{y}\left(x\right)=w_{0}+\overset{n}{\underset{i=1}{\sum }}w_{i}x_{i}+f\left(x\right)$$
where $w_{0}$ is a global bias, $w_{i}$ is the weight of the *i*-th feature, and f(x) is a multi-layer feedforward neural network that is used to model complex feature interactions. There are four components in f(x). (i) The embedding layer, where each feature is projected into a dense vector representation to obtain embedding vectors as Equation (8):(8)$$V_{x}=\left\{x_{1}v_{1},\ldots ,x_{n}v_{n}\right\}$$
where $v_{i}$ is the embedding vector for the *i*-th feature and $\underline{x}$ is the input feature vector. (ii) The bi-interaction layer, a pooling layer that converts $V_{x}$ obtained in the previous step to one vector as in Equation (9):(9)$$f_{Bi}\left(V_{x}\right)=\overset{n}{\underset{i=1}{\sum }}\overset{n}{\underset{j=i+1}{\sum }}x_{i}v_{i}⊙x_{j}v_{j}$$
where ⊙ denotes the element-wise product of two vectors, that is, ${\left(V_{i}⊙V_{j}\right)}_{k}=V_{ij}V_{ik}$. (iii) Hidden layers, a stack of fully connected layers that can learn higher-order interactions between features as in Equations (10)–(12):(10)$$z_{1}=\sigma_{1}\left(W_{1}f_{Bi}\left(V_{x}\right)+b_{1}\right)$$
(11)$$z_{2}=\sigma_{2}\left(W_{2}z_{1}+b_{2}\right)$$
……
(12)$$z_{L}=\sigma_{L}\left(W_{L}z_{L-1}+b_{L}\right)$$
where subscript *L* represents the number of hidden layers, and $W_{L}$, $b_{L}$, and $\sigma_{L}$ denote the weight matrix, bias vector, and activation function for the *L*-th layer, respectively. (iv) The prediction layer, the final prediction score is calculated based on the output vector of the last hidden layer $Z_{L}$ as in Equation (13):(13)$$f\left(x\right)=p^{T}Z_{L}$$
where *p* is the neuron weights of the prediction layer and superscript *T* indicates transpose operation. The formulation of NFM’s predictive model can be summarized as Equation (14): (14)$$\hat{y}\left(x\right)=w_{0}+\overset{n}{\underset{i=1}{\sum }}w_{i}x_{i}+p^{T}\sigma_{L}\left(W_{L}\left(\ldots \sigma_{1}\left(W_{1}f_{Bi}\left(V_{x}\right)+b_{1}\right)\ldots \right)+b_{L}\right)$$

The model hyperparameters for Task 1 are as follows: the size of the embedding layer is set to 64; the dropout ratio of the bi-interaction layer is set to 0. The number of neurons in the four hidden layers is 128 and the final prediction layer uses sigmoid as the activation function. The learning rate is set as 0.005 and other hyperparameters are set to default values.

Task 2 is a binary classification problem, i.e., identifying the types of DDIs, which is implemented by a four-layer DNN. The number of neurons in the four hidden layers is 512, 256, 128, and 64, respectively. The first three layers adopt ReLU as the activation function, while the last layer uses SoftMax. 

### 3.5. Baselines

In order to assess the performance of the current method, a comparison was performed with five baseline methods including random forest (RF) [53], KGNN [37], CNN-LSTM [54], KGE_NFM [35], and DDKG [55]. The RF model was taken from Scikit-learn [56], and DDIs were characterized by concatenating Morgan molecular fingerprints and KG embedding, the number of trees was 200, and other parameters were set as the default. Based on both a knowledge graph and a neural network, the KGNN is utilized to capture knowledge graph structure by mining relations in the KG. Similar to our embedding extraction method, the KGNN can effectively capture drugs and their potential neighborhood by mining the association relationship between drugs and their potential neighborhood in the KG. Drug pairs were represented using only embedding representations extracted from the high-dimensional and complex knowledge graph. CNN-LSTM combines CNN and LSTM layers to obtain important features from knowledge graph embedding. A CNN uses a convolutional filter to capture local relation values in drug characteristics, and the LSTM network can obtain global relations from features extracted by the CNN. Then using a global MPL, the most influential features are fed through another dropout layer into a fully connected layer and finally into the SoftMax layer to generate the probability distribution of the class. KGE_NFM utilizes DisMult to learn low-dimensional representations of diverse entities within KG and then integrates them using an NFMs. DDIs were characterized by embedding representation extracted by the DisMult algorithm and Morgan fingerprint splicing, which were finally input into the NFMs model for binary prediction. The DDKG is an attention-based KG representation learning framework, which utilizes the information of KGs for DDI predictions. A DDKG first initializes the drug representation using the embeddings obtained from the drug properties at the encoder-decoder layer, then learns the drug representation by recursively propagating and aggregating first-order adjacent information along the top-level network path determined by adjacent node embeddings and triple facts. Finally, the DDKG estimates the probability of DDIs with their characterization in an end-to-end manner. The optimal parameter settings were used in the baseline model experiments above.

### 3.6. Performance Evaluation

Based on the traditional 10-fold cross-validation (CV), area under the receiver operating characteristic curve (AUROC) [57], area under the precision–recall curve (AUPR), accuracy (Acc), sensitivity (Sen), specificity (Spe), precision (Pre), F1-score (F1) and Matthew’s correlation coefficient (MCC) were used to evaluate KGCN_NFM and the baseline models. In spite of the fact that the 10-fold CV can avoid the model suffering from overfitting in some cases, it may also produce overly optimistic results [58]. To make a realistic evaluation of the DDI prediction task, KGCN_NFM, and other baseline models were further evaluated on realistic scenarios proposed by pairwise disjoint CV (PW-CV) [59] and compared the performance with traditional CV. The PW-CV is a cold-start scenario designed to verify the ability to handle unseen nodes, where a drug in a drug pair contained in the test set is inaccessible in the training set. In addition, the 10-fold CV was performed for 10 repetitions and statistically averaged results were calculated to avoid the bias of data splitting.

### 3.7. Cytotoxicity Assays and Synergistic Effect

The cells were cultured on DMEM medium at 37 °C in a humidified atmosphere with 5% CO_2_. The cytotoxicity and drug combination effect of topotecan and dantron was evaluated through 3-[4,5-dimethylthiazol-2-yl]-2,5-diphenyltetrazolium bromide (MTT) assay against non-small cell lung cancer A549. The cancer cells were treated with the antineoplastic drugs (redissolved in DMSO) at a five-dose assay range from 12.5 to 200 μM for 48 h at 37 °C. MTT solution (10 μL, 5 mg/mL) in PBS buffer was added. After 4 h of incubation, 150 μL of DMSO was used to dissolve the formazan crystal formed in the wells for optical density determination at 570 nm by an enzyme-labeled instrument. The IC_50_ value was calculated by nonlinear regression analysis based on the GraphPad Prism [60]. By using Compusyn software (Version: 1.0), combination index (CI) values were calculated by using the Chou–Talalay method [61,62].

### 3.8. Apoptosis and Cell Cycle Analysis

The annexin V-FITC apoptosis detection kit (Beyotime, Shanghai, China) was used to analyze the apoptosis of A549 cells. Briefly, 1.5 × 10^5^ cells were plated in a 6-well plate, exposed with 5 µM topotecan and 10 µM dantron separately and together for 48 h. After that, the cells were digested with trypsin, centrifuged at 1000 rpm for 5 min, collected, and rinsed 3 times with precooled phosphate-buffered saline (PBS). Next, we resuspended the cells in 200 µL binding buffer and mixed 5 µL Annexin V-FITC with 10 µL propidium iodide (PI) staining solution. The cell apoptosis was examined by flow cytometry after incubation for 10 min. To assess the cell cycle, A549 cells were placed in a 6-well plate and exposed to 5 µM topotecan and 10 µM dantron separately and together. After 48 h, cells were permeabilized by 70% ethanol for 12 h and resuspended in a dyeing buffer. The cell cycle was analyzed by using a BD FACSCantoII (Becton, Dickinson and Company, Franklin Lake, NJ, USA) and data were analyzed using FlowJo (Version: 10.6.2, TreeStar, San Jose, CA, USA).

### 3.9. Molecular Docking

The structures of key proteins (AMPK, ID:3h4j; mTOR, ID:7ped; p53, ID:4d1m; sestrin, ID:5dj4; ULK1, ID:4wno) in the AMPK/mTOR pathway were downloaded from RCSB PDB (https://www.rcsb.org/ (accessed on 1 January 2023)). The 3D structures of small molecule drugs (topotecan: ZINC1611274; dantron: ZINC3860369) were retrieved from ZINC [63]. Molecular docking simulations of drug molecules were performed to verify that these two compounds may act together on different proteins in AMPK/mTOR pathway to exert synergistic antitumor effects. In addition, the docking program AutoDockTools (Version:1.5.6; for details, see Appendix A) was used for semi-flexible docking (i.e., the docking receptor protein is set as rigid and the docking ligand small molecule is considered as flexible). The size of the grid box was set to 126 Å × 126 Å × 126 Å with a grid spacing of 0.765 Å to enclose the active site of the AMPK/mTOR pathway receptors. All the other miscellaneous parameters were set to default. During the docking process, the Autodock program was utilized to calculate the binding energy between the receptor and the ligand. To find the optimal ligand–receptor binding, the genetic algorithm was adopted. For each protein, the number of the genetic algorithm is set to 50, and the population size is 150. Molecular docking analysis was evaluated based on the inhibition constant and docking energy, and the optimal docking conformation was retained to show a two-dimensional graph and three-dimensional diagram.

## 4. Conclusions

In this study, a framework called KGCN_NFM was proposed for discovering latent DDIs based on knowledge graphs and Morgan molecular fingerprints. The novel method has performed excellently in extensive investigations including model comparison, feature evaluation, ablation experiments, MTT assay, apoptosis applications, cell cycle analysis, and molecular docking. These results show that KGCN_NMF is a powerful DDI prediction tool with excellent performance because of three main reasons: (1) a high-quality dataset and reliable negative samples were selected rather than randomly selected from unlabeled samples; (2) the combination of drug structural information and knowledge graph-based embedding information contains rich domain information and semantic institutional features; (3) with the NFMs module, second-order feature interactions were further captured using feature intersection pooling to provide more information than concatenation.

In real situations, DDIs can be divided into many types, such as drug metabolism increased/decreased, the risk or severity of adverse effects increased/decreased, and the serum concentration increased/decreased. However, current research can only discriminate whether the type of interaction between the two drugs is increased or decreased. With the rapid development of various omics and the accumulation of relevant data, this problem will be further solved.

## Figures and Tables

**Figure 1 molecules-28-01490-f001:**
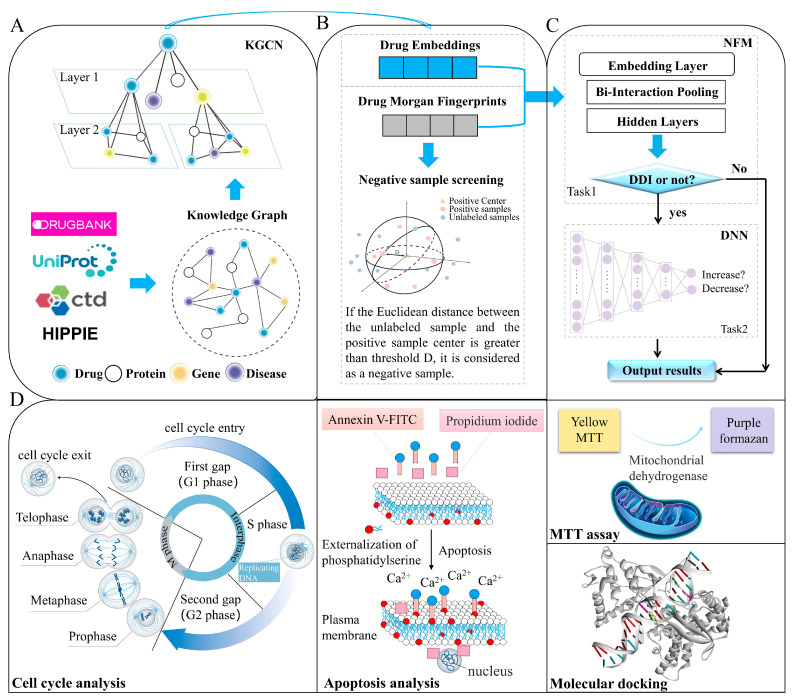
The schematic workflow of the current method. The framework mainly consists of four parts. (**A**) The KGCN extracts the embedding of entities and relations from the original input of DDIs data and related omics data. (**B**) Screen reliable negative sample based on Morgan fingerprints. (**C**) Model construct and performance evaluation. (**D**) Experimental verification through MTT assay, apoptosis experiment, cell cycle analysis, and molecular docking.

**Figure 2 molecules-28-01490-f002:**
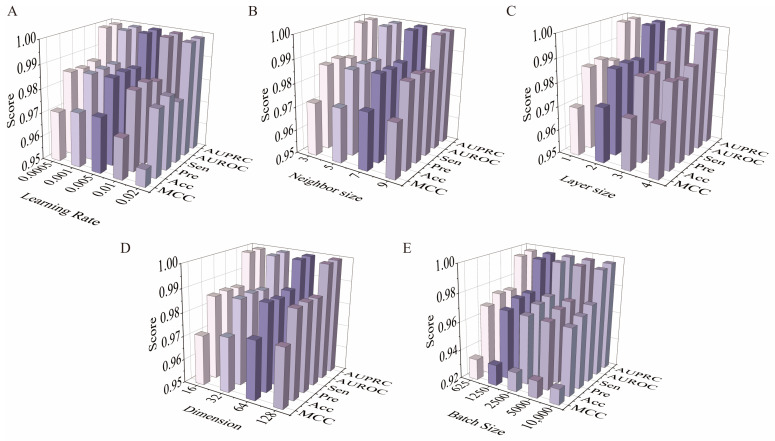
Optimization results of model parameters. (**A**) Effect of the parameter learning rate of the NFMs on model performance. (**B**) Effect of the neighbor size of the KGCN on model performance. (**C**) Effect of the number of layers of the KGCN on model performance. (**D**) Effect of the embedding dimension of the KGCN on model performance. (**E**) Effect of the batch size of the DNN on model performance.

**Figure 3 molecules-28-01490-f003:**
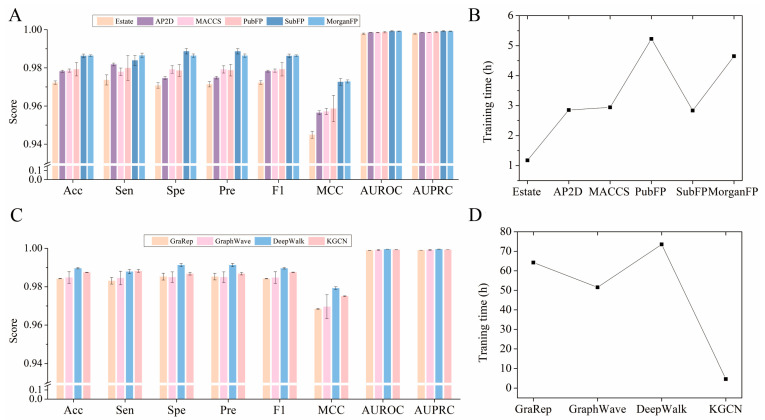
Effect of different fingerprints and embedding on model performance. (**A**) Effect of different fingerprints on model performance. (**B**) Training time required by models constructed with different molecular fingerprints. (**C**) Effect of different embedding on model performance. (**D**) Training time required by models constructed with different embedding.

**Figure 4 molecules-28-01490-f004:**
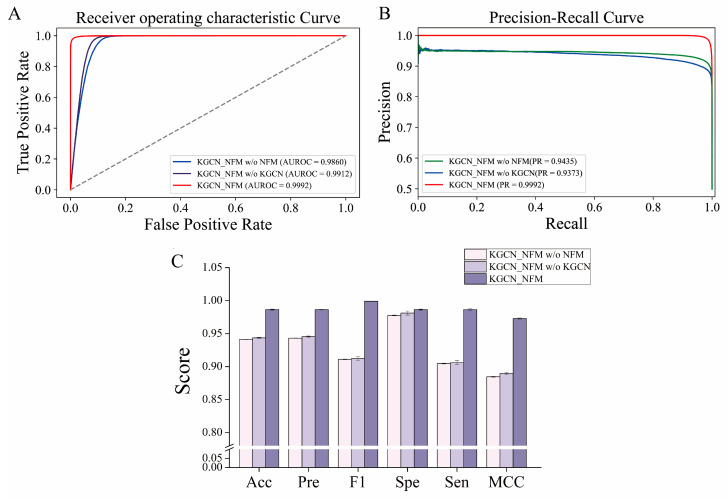
The impact of each component in the KGCN_NFM framework on the prediction performance. (**A**,**B**) represent the ROC and PR curves of the model with or without the removal of certain components, respectively. (**C**) represents the effect of different components on model performance.

**Figure 5 molecules-28-01490-f005:**
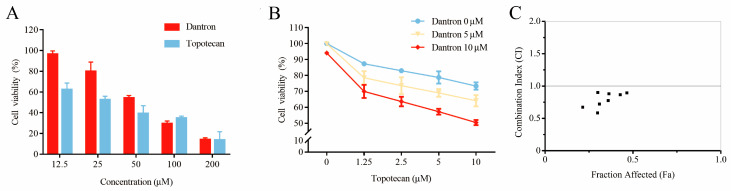
Potentiation of the cytotoxic action of topotecan in combination with dantron in A549 cells. (**A**) The cytotoxic effects of topotecan and dantron, respectively. (**B**) Curves of dose–response for combination treatment with topotecan and dantron. (**C**) Combination index (CI) vs. fraction affected (Fa) plot for the dose–response graphs. CI < 1, synergism.

**Figure 6 molecules-28-01490-f006:**
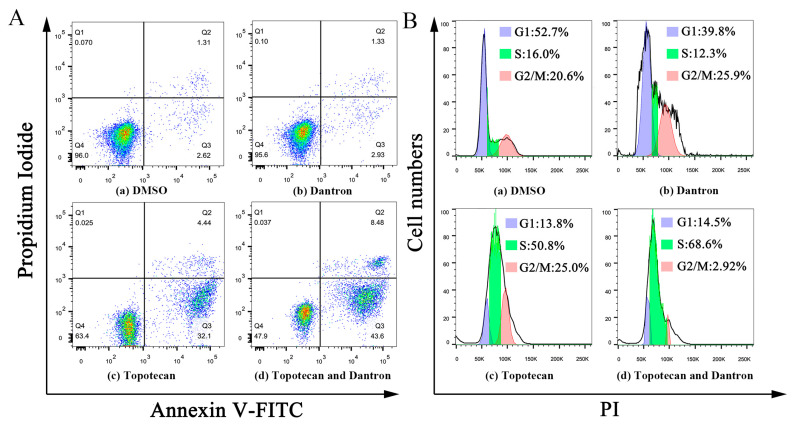
Effects of the combination of topotecan and dantron on cell apoptosis and cell cycle in A549 cells. (**A**) represents the effect of the combination of the two drugs on apoptosis. (**B**) represents the effect of the combination of the two on the cell cycle.

**Figure 7 molecules-28-01490-f007:**
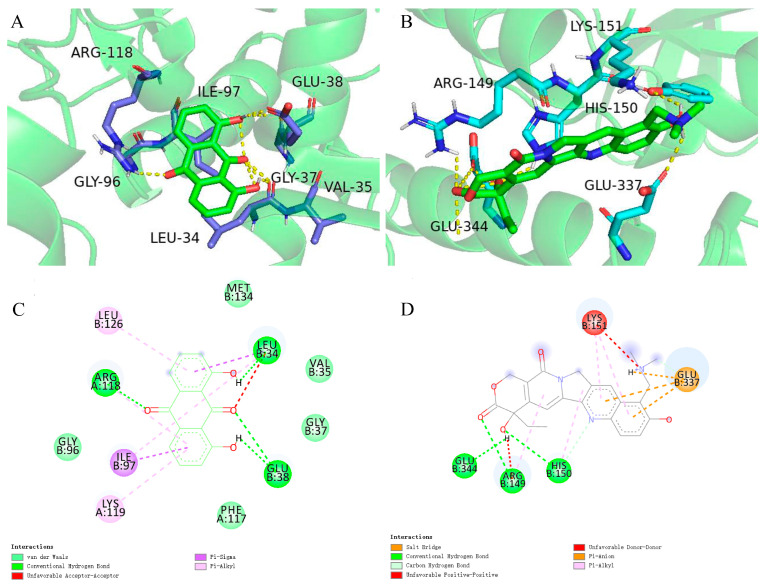
Molecular docking results. (**A**) represents the structure of mTOR (RCSB PDB:7ped) with bound topotecan. (**B**) represents the molecular docking results of AMPK (RCSB PDB:3h4j) and dantron. (**C**) Schematic representation of the 2D interactions between dantron and mTOR. (**D**) Schematic representation of the 2D interactions between topotecan and the AMPK.

**Table 1 molecules-28-01490-t001:** Impact of the different negative sample selecting thresholds on the 8 evaluation metrics (mean ± RSD (%)).

Threshold	Acc (%)	Sen (%)	Spe (%)	Pre (%)	F1	MCC	AUC	AUPR
UN	96.20 ± 0.13	96.97 ± 0.23	95.42 ± 0.33	95.50 ± 0.30	0.9622 ± 0.13	0.9241 ± 0.27	0.9908 ± 0.06	0.9882 ± 0.08
0.6 AED	96.40 ± 0.09	97.07 ± 0.29	95.74 ± 0.18	95.80 ± 0.16	0.9643 ± 0.09	0.9282 ± 0.19	0.9917 ± 0.02	0.9895 ± 0.03
0.8 AED	96.46 ± 0.05	97.18 ± 0.22	95.75 ± 0.17	95.81 ± 0.15	0.9649 ± 0.05	0.9295 ± 0.11	0.9919 ± 0.01	0.9898 ± 0.02
1.0 AED	97.06 ± 0.06	97.50 ± 0.16	96.62 ± 0.12	96.65 ± 0.11	0.9707 ± 0.06	0.9413 ± 0.11	0.9954 ± 0.01	0.9949 ± 0.01
1.2 AED	98.64 ± 0.04	98.65 ± 0.12	98.64 ± 0.09	98.64 ± 0.09	0.9864 ± 0.04	0.9729 ± 0.08	0.9992 ± 0.01	0.9992 ± 0.01

AED: average Euclidean distance, nAED is defined as n × AED, which is the threshold with different distances from unlabeled samples to the cluster center. UN: randomly select negative samples.

**Table 2 molecules-28-01490-t002:** The performance of different methods on the benchmark dataset and KEGG-DDI dataset (mean ± RSD (%)).

Datasets	Methods	Acc (%)	Sen (%)	Spe (%)	Pre (%)	F1	MCC	AUC	AUPR
benchmark dataset	RF	89.50 ± 0.04	93.63 ± 0.01	85.37 ± 0.07	86.48 ± 0.06	0.8992 ± 0.04	0.7927 ± 0.08	0.9393 ± 0.02	0.8914 ± 0.04
	KGNN	97.09 ± 0.58	98.56 ± 0.20	95.63 ± 1.03	95.79 ± 0.93	0.9714 ± 0.55	0.9424 ± 1.15	0.9924 ± 0.38	0.9911 ± 0.34
	CNN-LSTM	97.55 ± 0.96	97.95 ± 0.66	97.14 ± 2.06	97.21 ± 1.97	0.9756 ± 0.93	0.9512 ± 1.94	0.9958 ± 0.31	0.9950 ± 0.38
	KGE_NFM	96.76 ± 0.05	97.83 ± 0.33	95.70 ± 0.26	95.79 ± 0.23	0.9679 ± 0.06	0.9356 ± 0.11	0.9929 ± 0.02	0.9912 ± 0.02
	DDKG	87.94 ± 0.82	96.96 ± 1.99	78.52 ± 1.35	82.46 ± 0.91	0.8911 ± 1.05	0.7707 ± 1.25	0.9217 ± 1.13	0.8920 ± 2.26
	Our (Task 1)	98.75 ± 0.01	98.83 ± 0.08	98.68 ± 0.07	98.68 ± 0.07	0.9875 ± 0.01	0.9751 ± 0.02	0.9993 ± 0.01	0.9994 ± 0.01
	Our (Task 2)	96.84 ± 0.01	97.42 ± 0.04	96.00 ± 0.07	97.32 ± 0.04	0.9737 ± 0.01	0.9344 ± 0.01	0.9955 ± 0.01	0.9970 ± 0.01
KEGG-DDI	RF	90.51 ± 0.11	93.19 ± 0.14	87.83 ± 0.18	88.45 ± 0.15	0.9075 ± 0.11	0.8113 ± 0.25	0.9676 ± 0.04	0.9664 ± 0.05
	KGNN	87.45 ± 0.38	91.98 ± 1.27	82.91 ± 0.84	84.39 ± 0.44	0.8800 ± 0.48	0.7525 ± 0.98	0.9348 ± 0.33	0.9108 ± 0.37
	CNN-LSTM	98.58 ± 0.03	96.25 ± 0.47	94.93 ± 0.97	95.04 ± 0.89	0.9563 ± 0.61	0.9716 ± 0.05	0.9882 ± 0.26	0.9862 ± 0.27
	KGE_NFM	84.22 ± 0.62	86.82 ± 1.69	81.57 ± 1.21	82.91 ± 0.76	0.8468 ± 0.80	0.6878 ± 1.49	0.9203 ± 0.50	0.9090 ± 0.50
	DDKG	83.07 ± 0.61	84.48 ± 3.49	81.66 ± 2.65	82.13 ± 1.47	0.8324 ± 1.09	0.6624 ± 1.81	0.9191 ± 0.61	0.9202 ± 0.65
	Our	98.85 ± 0.03	98.36 ± 0.05	98.80 ± 0.04	98.79 ± 0.04	0.9857 ± 0.03	0.9716 ± 0.05	0.9987 ± 0.01	0.9988 ± 0.01

**Table 3 molecules-28-01490-t003:** The performance of different methods on traditional CV and PW-CV scenarios.

Methods	Traditional CV							PW-CV						
	Acc (%)	Sen (%)	Spe (%)	Pre (%)	F1	AUC	AUPR	Acc (%)	Sen (%)	Spe (%)	Pre (%)	F1	AUC	AUPR
RF	89.50	90.63	85.37	86.48	0.8992	0.9393	0.8914	67.10	45.72	89.09	81.16	0.5840	0.8006	0.7851
KGNN	97.09	98.56	95.63	95.79	0.9714	0.9924	0.9911	38.96	55.02	22.41	42.08	0.4760	0.4704	0.5861
CNN-LSTM	97.55	97.95	97.14	97.21	0.9756	0.9958	0.9950	51.39	74.63	27.66	52.08	0.5247	0.6115	0.6227
KGE_NFM	96.76	97.83	95.70	95.79	0.9679	0.9929	0.9912	72.56	56.69	88.43	84.32	0.6727	0.7256	0.8148
DDKG	87.94	96.96	78.52	82.46	0.8911	0.9217	0.8920	86.74	97.13	76.03	80.73	0.8815	0.8925	0.8333
Our	98.75	98.83	98.68	98.68	0.9875	0.9993	0.9914	83.18	82.45	83.95	84.39	0.8324	0.9083	0.8775

**Table 4 molecules-28-01490-t004:** The ten top-ranked drugs with interactions with topotecan.

DrugBank ID	Name	Structure	Evidence
DB14783	Diroximel fumarate	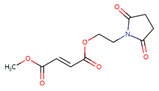	Drugs.com
DB11793	Niraparib	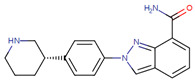	Drugs.com
DB09269	Phenylacetic acid	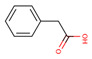	PMID: 33218116
DB15091	Upadacitinib	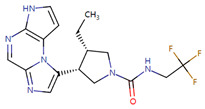	Drugs.com
DB11942	Selinexor	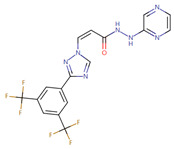	Drugs.com
DB04816	Dantron	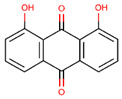	unconfirmed
DB03419	Uracil	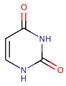	PMID: 22543158
DB01208	Sparfloxacin	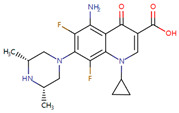	Drugs.com
DB01059	Norfloxacin	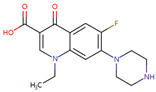	Drugs.com
DB02690	NU1025	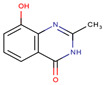	PMID:10914735

## Data Availability

The source codes and data are available at https://github.com/zhangjing9965/KGCN_NFM (accessed on 1 January 2023).

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
