# Peer review of "A Knowledge-Graph-Based Multimodal Deep Learning Framework for Identifying Drug–Drug Interactions"

_molecules, 2023, doi:10.3390/molecules28031490_

Round 1
Reviewer 1 Report
In the current study, a hybrid deep learning method has been developed by combining KGCN and NFM methods. Through the benchmark study and case study, the newly developed outperforms other methods or separated components of itself. I think the study was well conducted, but before the publication, the following concerns have to be addressed.
1. The molecular docking subsection is missing in the methods section (although a step-by-step guide has been introduced in SI part 2), where the details of the molecular docking simulation need to be described in depth, the software used, the number of gestures that were evaluated, the position of the docking pocket and how the docking pocket was selected, etc.
2. I encourage that the authors could make their machine learning code public available, either to Github or to Gitlab, so that everyone who has interest on the method could easily verify the implementation. The version information of database and RDkit need to be specified as well since they really change a lot with time evolves.
3. The authors may want to emphasize how their novel method could help the drug development in reality. For example, if a new drag targeting a particular disease has been developed and been shown efficacy, how to predict the potential disadvantages of the developed drug molecule when working together with other drugs?
Reviewer 2 Report
1. This paper uses the KGCN knowledge map convolutional neural network. The KGCN model is sampled from the neighbors of each entity. Please demonstrate the impact of using a fixed-size collection with a complete neighbor collection.
2. Please add the construction process of the NFM model.
3. Please explain if the NFM model is calculated with the same vector when the features cross with other features. Is there a different degree of cross-crossing between different characteristics?
4. Please express the comparison process of the five baseline methods in detail.
5. Please add the quantitative results of cross-verification in the 2.6 performance assessment.
Round 2
Reviewer 2 Report
Accept.